# Condition Monitoring of Railway Tracks from Car-Body Vibration Using a Machine Learning Technique

**Hitoshi Tsunashima** †

Department of Mechanical Engineering, Nihon University, Chiba 275-8575, Japan;
tsunashima.hitoshi@nihon-u.ac.jp; Tel.: +81-47-474-2339
† Current Address: 1-2-1 Izumi-cho, Narashino-shi, Chiba, Japan.

**Abstract:** A track condition monitoring system that uses a compact on-board sensing device has been developed and applied for track condition monitoring of regional railway lines in Japan. Monitoring examples show that the system is effective for regional railway operators. A classifier for track faults has been developed to detect track fault automatically. Simulation studies using SIMPACK and field tests were carried out to detect and isolate the track faults from car-body vibration. The results show that the feature of track faults is extracted from car-body vibration and classified from proposed feature space using machine learning techniques.

**Keywords:** railway; condition monitoring; fault detection; preventive maintenance; machine learning

---

## 1. Introduction

With the recent development of sensors and information technology (IT), conditions in railway facilities can be monitored continually by using sensors installed in the rolling stock and in areas adjacent to the track. This, in turn, has spurred the interest in using this type of monitoring to create maintenance plans or schedule condition-based maintenance when the track conditions indicate deterioration.

The condition of railway tracks is an important factor in ensuring the safe operation of trains. To ensure that railway travel is safe and comfortable, it is necessary to maintain and manage tracks properly; this includes preventive maintenance. Further, it is desirable to monitor tracks frequently. The deformations that occur in tracks, referred to as track irregularities, are closely connected to the riding quality and safety of railway vehicles. Urban railway operators use track inspection cars to verify track irregularities.

Track condition monitoring systems that observe car-body vibrations in in-service vehicles have already been developed for regional railways and have exhibited the capability to oversee track conditions [1]. Additionally, recent studies proposed employing a Kalman filter to predict track irregularities from the acceleration measurements of car-body vibrations of railway vehicles [2].

Moreover, on regional railway lines, many operators are unable to implement adequate inspections owing to problems such as cost and personnel constraints. Hence, low-cost monitoring systems have been developed based on on-board sensing devices and global navigation satellite system (GNSS) data. In this system, the sensing device is installed in an in-service vehicle to measure car-body vibrations. The diagnosis is performed by rating the track conditions based on the measurement data. Presently, however, the rating needs to be calculated manually. It is difficult to address a large volume of automatically collected data. To enable the efficient computational processing of collected data, the diagnosis and prediction of track conditions must be automated.

In the present study, we adopted machine learning techniques and developed an algorithm to automatically diagnose track conditions from car-body vibrations to automatically detect track irregularities and identify them by type. We subsequently performed a simulation to verify the effectiveness of the developed algorithm. In addition, we examined the possibility of diagnosing track conditions for an actual regional railway line.

## 2. Trends of Track Condition Monitoring

The monitoring of railway track geometry from an in-service vehicle became increasingly attractive over the past decade [3]. Track geometry measurement systems using in-service vehicles have been developed worldwide, although they are still in the developing stage. The repeated verification of the same track provides an opportunity to record track geometry degradation. The obtained information is fed back to the track maintenance section to implement necessary steps.

Several types of track faults are detected by measuring the acceleration of bogies. Weston et al. demonstrated track irregularity monitoring by using bogie-mounted sensors [4,5]. Alfi et al. proposed a technique to estimate long wavelength track irregularities from on-board measurements [6]. If track faults are detected using car-body mounted sensors, it is easier to perform the condition monitoring of track irregularities. The distinctive signal of track faults is hidden in car-body vibrations, and thus signal processing of the acceleration measured in-cabin is necessary to detect the track faults. Tsunashima et al. demonstrated the possibility of estimating the track geometry of Shinkansen tracks using only car-body motions [7,8]. A Kalman filter was applied to estimate the track irregularity from the car-body motions.

Tsunashima et al. developed a system that attempted to identify vertical and lateral track irregularities by using accelerometers placed on the car-body of in-service vehicles [9]. Furthermore, the system provides a function to listen for corrugation faults by using an acoustic sensor (a microphone). An approach was tested to detect squats by listening to rolling noise using microphones [10]. Odashima et al. demonstrated the possibility of estimating the track irregularities of conventional railway tracks by using only car-body acceleration [2]. The Kalman filter-based estimation technique was proposed for conventional railways and evaluated from data obtained with the track condition monitoring system developed by Tsunashima et al. [1,11].

## 3. Development of Track Condition Monitoring System

### 3.1. System Overview

Figure 1 presents an overview of the track condition monitoring system. The system in an in-service vehicle consists of a compact on-board sensing device for measuring car-body vibrations and diagnosis software for identifying track conditions from the data.

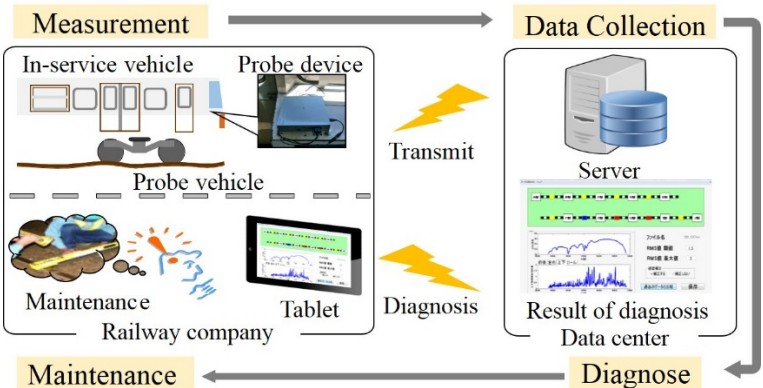

**Figure 1.** Track condition monitoring system.

The data are transmitted to a server via a wireless phone connection and examined by using track condition diagnosis software. The results of the diagnosis are fed back to the infrastructure providers where they are used to help create maintenance plans, select work locations, and assign work priorities.

### 3.2. On-Board Sensing Device

Figure 2 shows a photograph of the compact on-board sensing device used in the track condition monitoring system [1,11]. The device consists of an accelerometer and rate gyroscope for detecting track errors, a global navigation satellite system (GNSS) receiver for detecting the train location and speed, and a sensor interface for feeding the signals from the sensors to the computer.

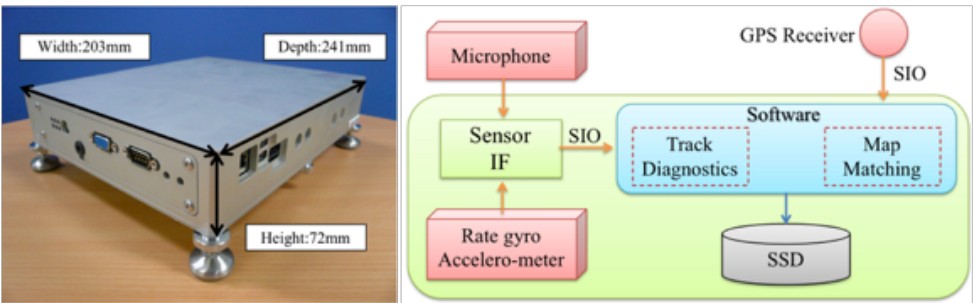

**Figure 2.** On-board sensing device.

The system is battery-powered and conducts continuous measurements for up to six hours, and thus measurements are simply obtained by placing it in a car. Measurements are constantly obtained if it is powered by a source on the car. Measurement data are stored into an on-board storage device. Furthermore, the device is equipped with a function to automatically transmit measured data from the on-board storage device to a server via a mobile phone network. The device is further equipped with a microphone, and thus, it is also possible to "listen" for corrugation and diagnose the condition.

The measured data obtained from the measurement device are transmitted to the analysis centre either via a mobile phone network or via writing to external media. The diagnostic results produced by the track monitoring software are used to provide feedback to railway operators through online channels via smartphones or tablet computers. Railway operators use the information to establish the track maintenance priorities, and thereby facilitate the maintenance planning and work.

Additionally, the real-time measurement of vehicle vibrations during in-service operation allows for rapid response, such as emergency track inspection and maintenance, in situations in which vehicle vibration observations detect irregularly large deviations from standard control values. Thus, the use of the monitoring system to perform continuous monitoring of vehicle vibrations allows early detection of deterioration or other track irregularities, and thereby enables infrastructure providers to conduct effective maintenance work.

## 4. Track Fault Detection Using Machine Learning

### 4.1. Track Irregularities

Track irregularity inspection includes vertical rail profile, lateral alignment, gauge, cross-level, twist and dipped joint, and corrugation as shown in Figure 3. In the developed monitoring system, the detection of track faults for vertical, lateral alignment, and cross-level is possible by using an on-board sensing device.

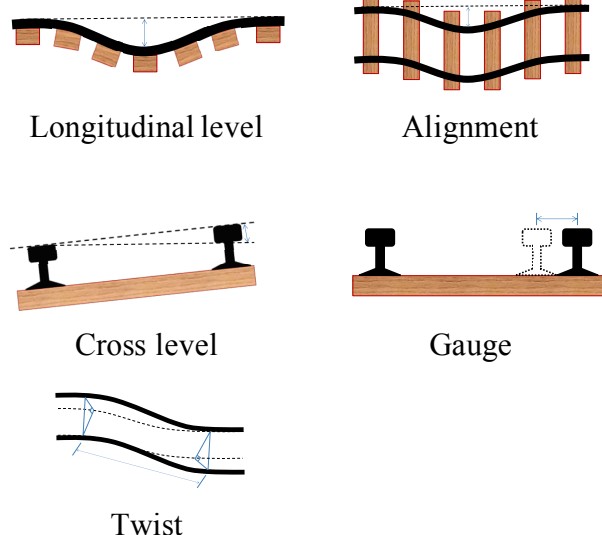

**Figure 3.** Track irregularities.

Track irregularities cause vehicle vibrations that degrade rider comfort and also increase the risk of derailments. Thus, they are among the most important items that are monitored. Vehicle vibrations are strongly correlated with track irregularities, and thus the magnitude of vehicle vibrations is an effective means for assessing general track condition trends. Thus, sections of track over which high levels of vehicle vibrational acceleration are detected can be considered as indicative of degraded track conditions.

### 4.2. Multibody Dynamics Simulation

To analyse the effects of railway track deterioration on car-body vibrations, we performed a railway vehicle travel simulation using SIMPACK, a multibody dynamics software package. The model used for the simulation was a single conventional railway vehicle, the parameters for which are shown in Table 1. Figure 4 shows the model used in the simulation.

**Table 1.** Vehicle parameters.

| Description | Unit | Value |
|---|---|---|
| Car-body mass | kg | 25,000 |
| Bogie mass | kg | 3100 |
| Wheelset mass | kg | 1500 |
| Car-body inertia | $kgm^2$ | 84,100 |
| Bogie inertia | $kgm^2$ | 1734.75 |
| Wheelset inertia | $kgm^2$ | 735 |
| Car-body base | m | 14.1 |
| Wheel base | m | 2.1 |
| Primary suspension vertical stiffness | kN/m | 2120 |
| Secondary suspension vertical stiffness | kN/m | 200 |
| Primary suspension lateral stiffness | kN/m | 5590 |
| Secondary suspension lateral stiffness | kN/m | 131.3 |
| Primary suspension vertical damping | kNs/m | 78.4 |
| Secondary suspension vertical damping | kNs/m | 48 |
| Primary suspension lateral damping | kNs/m | 78.4 |
| Secondary suspension lateral stiffness | kNs/m | 58.8 |

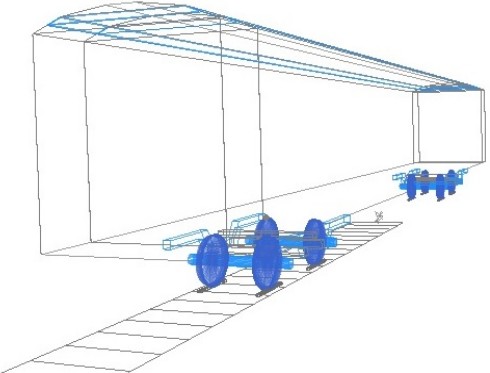

**Figure 4.** Vehicle model.

The model consists of seven rigid bodies, one car body, two bogies, and four wheelsets, each having six degrees of freedom. Thus, 42 degrees of freedom exist in the model. It is necessary to connect each rigid body with an appropriate joint such that the vehicle exhibits a realistic motion. One car-body, two bogies, and four wheelsets were connected by spring and damper elements. These spring and damper elements were set with spring constants and damping coefficients in three axial directions.

Using this vehicle model, we calculated the vertical acceleration, lateral acceleration, and roll rate above the centre of the front bogie at the car body. This simulation study is carried out based on the straight track. Alignment irregularity in curved track is not considered in this paper but should be considered in the next work.

### 4.3. Feature Extraction of Track Faults from Car-Body Vibration

Track irregularities can be used as an index for managing track conditions. The types of track irregularities include longitudinal level irregularities that indicate vertical rail displacement, alignment irregularities that indicate lateral rail displacement, and cross level irregularities that indicate a difference in the elevation of the left and right rails. Because a high degree of correlation exists between these track irregularities and car-body vibrations, the magnitude of car-body vibrations can be an effective method to determine track conditions [12,13]. The track-condition monitoring system can automatically collect the vertical and lateral accelerations and the roll rate of the car body during travel to detect any longitudinal level irregularities, alignment irregularities, and cross level irregularities.

To identify different types of track faults by using machine-learning techniques, it is necessary to extract the features of each type of track irregularity from the car-body vibrations that occur when travelling on a faulty track. The data obtained by learning the extracted features can be used to construct a training model. Subsequently, if unknown car-body vibration data are input to the model, the type of track irregularity and its degradation level can be identified.

As shown in Figure 5, we created a baseline track geometry for an assumed regional railway line and a track geometry with increased track irregularities. We subsequently performed a travel simulation by using a track for which all the longitudinal level irregularities, alignment irregularities, and cross level irregularities were within the normal range, and also by using a track for which one type of irregularity was abnormal. Degraded track is generated by the power spectral density of track where the standard deviation becomes twice as large as the normal track. The travelling distance was 1000 m and the travelling speed was 60 km/h.

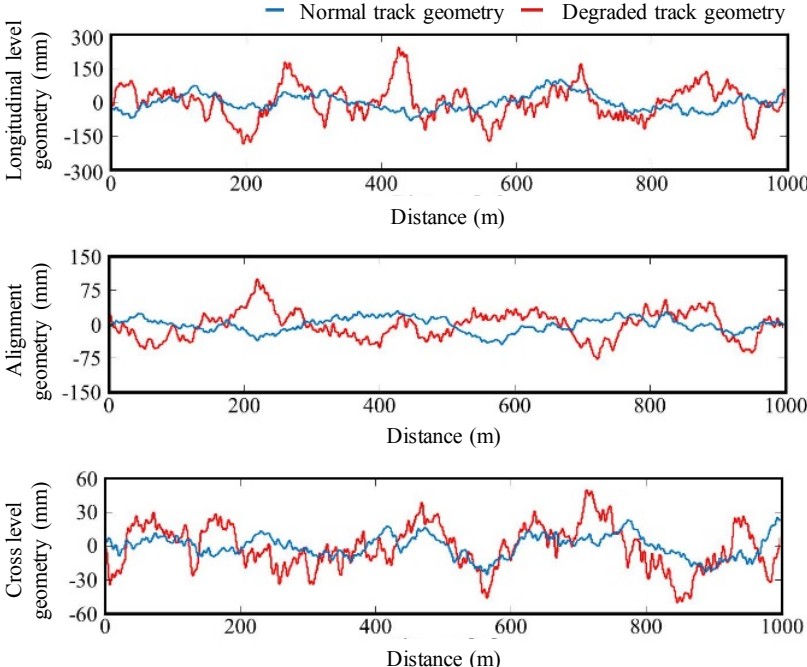

**Figure 5.** Baseline and degraded track geometry.

The acceleration root mean square (RMS) is closely linked to the general health of the running gear and track [12]. Therefore, the acceleration RMS can be used as the primary index to predict the track condition. First, we need to obtain localised RMS values over short time intervals with a small window size to obtain the relationship between track condition and position. The RMS value may be obtained from measured values over a short time segment according to

$$x_{rms}^i = \sqrt{\frac{1}{N} \sum_{j=i}^{i+N-1} x_j^2},$$
(1)

where $x_j$ is the measured vertical acceleration, lateral acceleration or roll angular velocity of car-body and $N$ denotes the sliding window size.

Track irregularities include longitudinal level irregularities, alignment irregularities, and cross level irregularities. In order to assess the irregularities, we determine the RMS values of vertical acceleration, lateral acceleration, and roll rate measured by the compact-sized on-board sensing device. We compute RMS values by using $N = 4$ and a sampling frequency of 82 Hz. The number $N$ should be optimised based on the analysis of collected measurement data as the future work.

Figure 6 shows the results of car-body vibration calculations, which indicate that longitudinal level and alignment irregularities influence the vertical and lateral accelerations, respectively, while cross level irregularities influence both the lateral acceleration and roll rate. The maximum RMS values for the vertical acceleration, lateral acceleration, and roll rate were calculated for each 10 m section. The section length 10 m is decided based on the consideration of GNSS error.

Figure 7 shows a feature space created by plotting these values in three dimensions. As shown, four clusters are present, representing a normal track, and abnormal tracks with longitudinal level, alignment, and cross level irregularities. It is obvious that such a feature space could be used as training data for machine learning to enable track conditions to be identified.

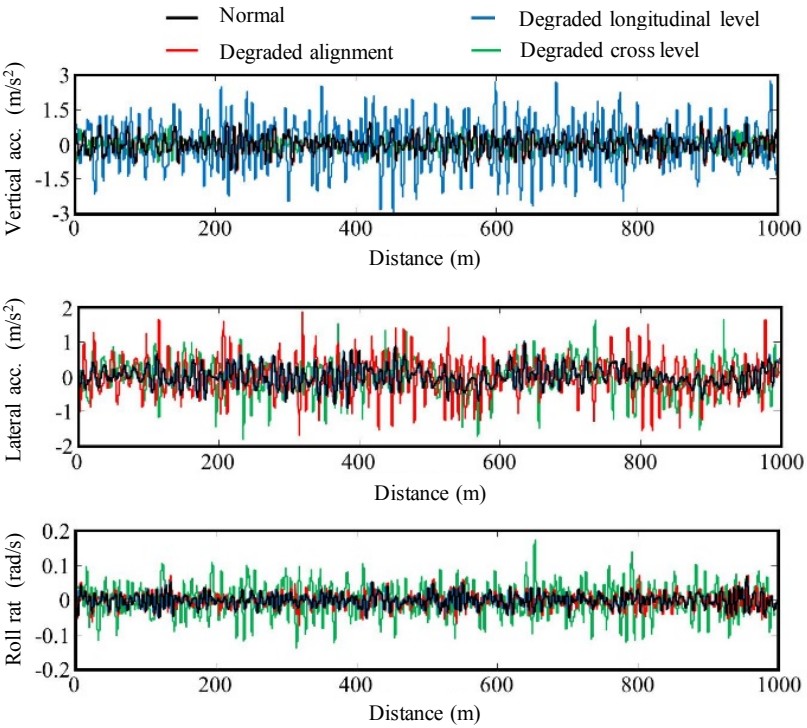

**Figure 6.** Calculated car-body vibration.

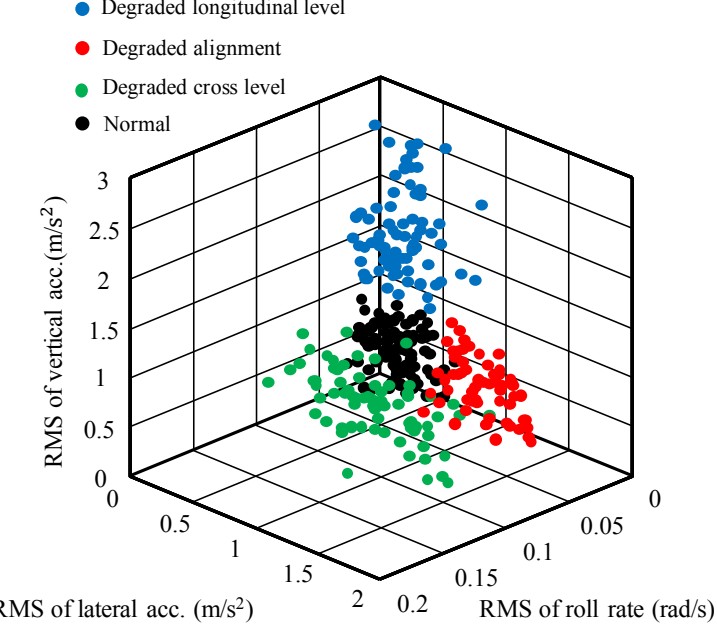

**Figure 7.** Feature space of car-body vibration RMS.

*4.4. Detection of Track Faults Using Machine Learning*

In this study, we used a support vector machine (SVM), which is a type of machine-learning algorithm created to solve two-class classification problems. An SVM identifies the most distant points ("maximum margin") between two groups and draws a boundary at the centre point between them. A feature of the SVM is that maximising this separation (margin) results in a high degree of generalisation. Utilising the SVM, we classified track conditions by using the following procedure.

First, we classify the obtained data into two classes, "faulty track with longitudinal-level irregularity" or "not." Next, we classify the data into "faulty track with alignment irregularity" or "not." Subsequently, we classify the data into "faulty track with cross-level irregularity" or "not." Finally, if the data are not classified in the faulty data group, the track is diagnosed as normal. By using the procedure above, the track is diagnosed for every 10 m section with the algorithm.

We have not considered for optimising parameters in SVM [14] in this study, but it should be considered in the future work.

## 5. Diagnosis Simulation

### 5.1. Generation of Car-Body Vibration Data

We performed a vehicle travel simulation using SIMPACK and calculated the car-body vibration when the vehicle was traveling on tracks with different irregularities in some sections. The obtained data were used to detect track faults using the developed algorithm.

The track geometry used for the simulation differed from that shown in Figure 8. The track faults were modelled by increasing the reference track irregularities by factors of 1.5 and 2.0 times and added to the reference track geometry. Figure 9 shows the modelled track faults. The track geometry used in the simulation was generated by adding the track fault in Figure 8 to the reference track geometry. The total length of the track geometry was 5000 m. Furthermore, 100 faults of longitudinal level, alignment, and cross level irregularities were randamly provided in different places.

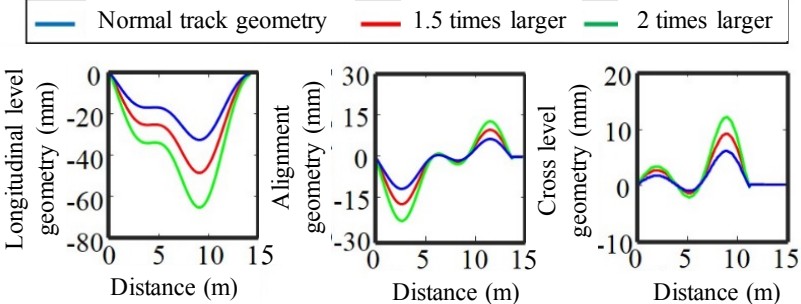

**Figure 8.** Track fault.

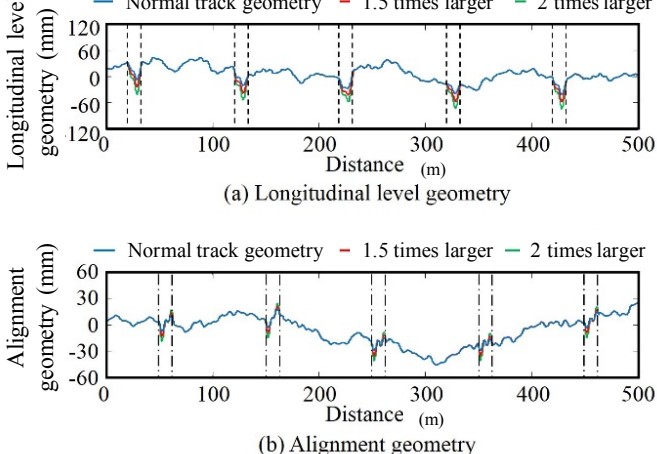

**Figure 9.** *Cont.*

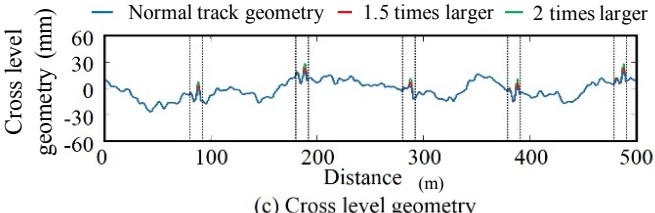

**Figure 9.** Degraded track geometry in some sections. (**a**) Longitudinal level geometry. (**b**) Alignment geometry. (**c**) Cross level geometry.

Figure 10 shows the calculation result of the car-body vibrations when a vehicle travels the track geometry shown in Figure 9. As shown in Figure 10, only the vertical or lateral acceleration increases when the longitudinal level irregularity or the alignment irregularity increase. As shown, both the lateral acceleration and roll rate increase when the cross level irregularity increases.

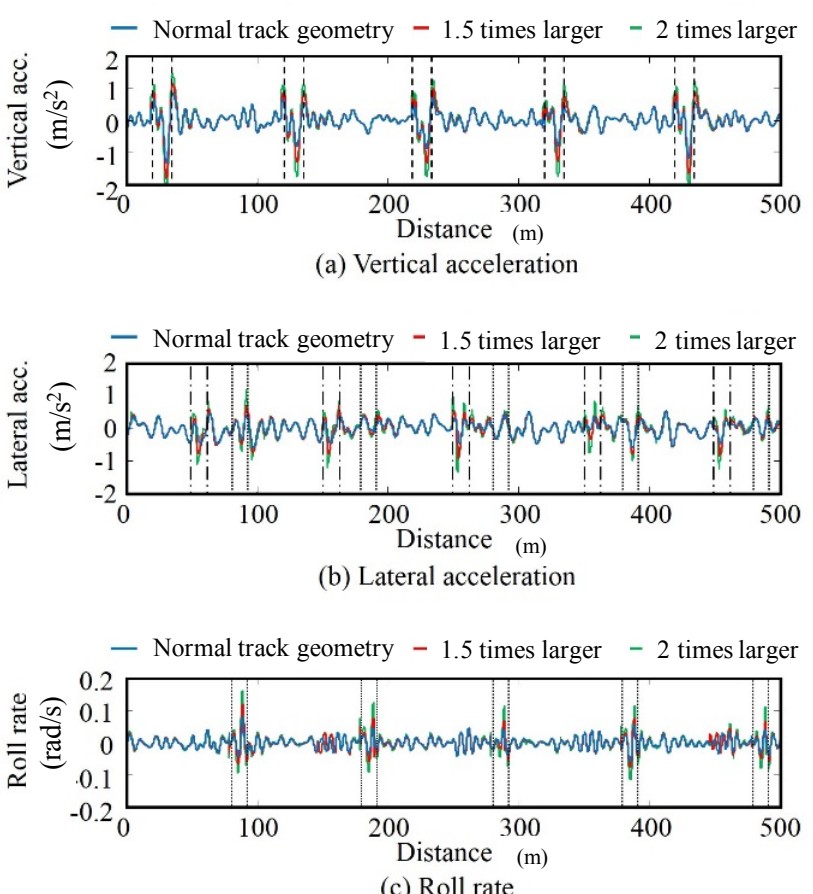

**Figure 10.** Calculated car-body vibration by using Figure 9. (**a**) Vertical acceleration. (**b**) Lateral acceleration. (**c**) Roll rate.

### 5.2. Fault Detection Results

We examined the effectiveness of the developed algorithm by detecting the track fault from the simulated car-body vibration data. For the training data, we used the feature space as shown in Figure 6. The detection results were divided into four classes: normal track, faulty track with longitudinal-level irregularity, faulty track with alignment irregularity, and faulty track with cross-level irregularity. The validity of the detection was evaluated for every 100 faults of longitudinal

level, alignment, and cross level irregularities in 5000 m track lengths. Figure 11 shows the results of detection by using the algorithm.

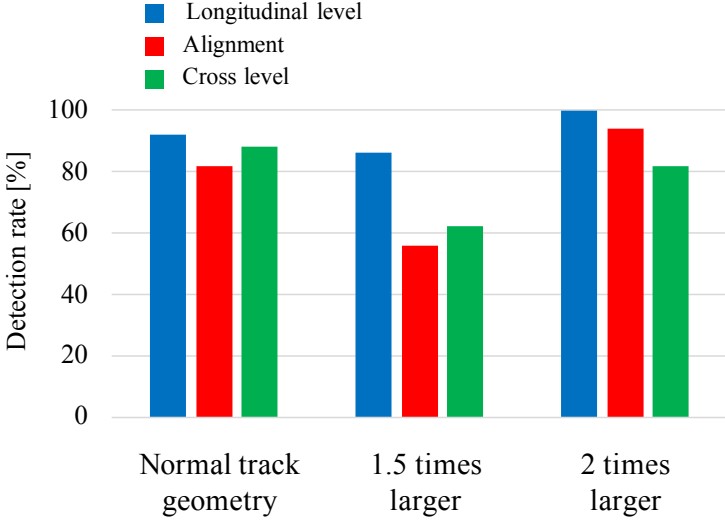

**Figure 11.** Detection results.

As shown in Figure 11, the developed algorithm detected the track fault with an accuracy of 80 percent or more for normal and highly degraded tracks. Clearly, the longitudinal level irregularity can be detected with high accuracy for normal, intermediate, and highly degraded tracks. Meanwhile, the detection accuracy of the alignment and cross level irregularities are lower than that for the longitudinal level irregularity, especially for the intermediately degraded track. However, it is noted that most of the track faults in the regional railway are longitudinal level irregularities. Therefore, the developed diagnosis algorithm is considered effective for regional railway lines.

The effect of vehicle faults such as wheel flat will appear on the car-body vibration. It is possible to identify the possibility of the faulty train by looking at the measurement data of different places.

## 6. Application for Regional Railway Line

As an initial test of track fault detection using machine learning, we created clusters of simple features from the car-body vibration data for a regional railway line. To measure the vibration, we used an on-board sensing device comprising an accelerometer, a gyroscope, and a GNSS receiver as shown in Figure 12.

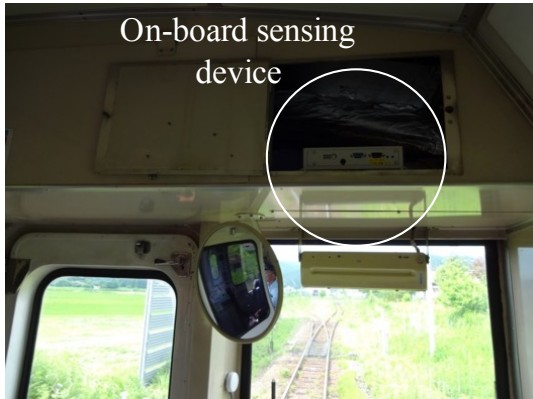

**Figure 12.** Measurement of car-body vibration in a regional railway line.

The training data were measured in January, February, May, June, July, and November 2016 for sections of the line where repairs for longitudinal level irregularities were performed in October 2016. Clusters of abnormal track data were created based on the following conditions:

- Longitudinal level irregularity: vertical acceleration $\geq 1.0\,\mathrm{m/s^2}$
- Alignment irregularity: lateral acceleration $\geq 1.1\,\mathrm{m/s^2}$ and roll rate $\leq 0.05\,\mathrm{rad/s}$
- Cross level irregularity: roll rate $\geq 0.05\,\mathrm{rad/s}$

Figure 13 shows the feature space of car-body vibration in a regional railway line. The normal cluster was constructed from car-body vibration data measured in November 2016, immediately after longitudinal level irregularities were repaired. The algorithm detects the location and type of track fault for every 10 m of track. Using the diagnosis algorithm, we diagnosed the track condition in May 2018.

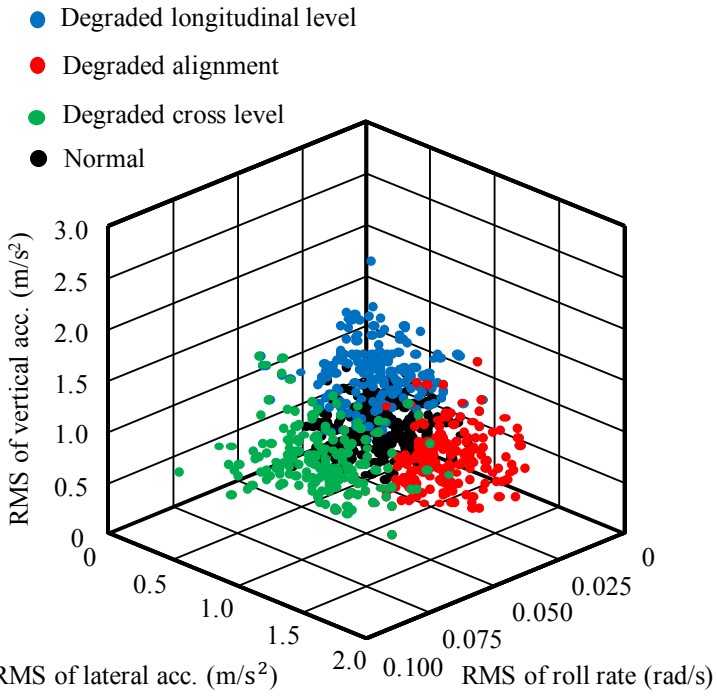

**Figure 13.** Feature space of car-body vibration in a regional railway line.

Figure 14 shows the measured RMS values for the car-body vibrations and the results obtained by the detection algorithm (colour map in the upper part of each figure). Locations where the RMS values for vertical acceleration, lateral acceleration, and roll rate are large are successfully diagnosed as longitudinal level, alignment, and cross level irregularities, respectively.

As shown, multiple track faults were detected in location from 25.8 km to 25.9 km. The results indicated that combined faults can be detected by the developed algorithm. Therefore, we can conclude that the proposed method is effective for identifying the location and type of track fault in a regional railway line.

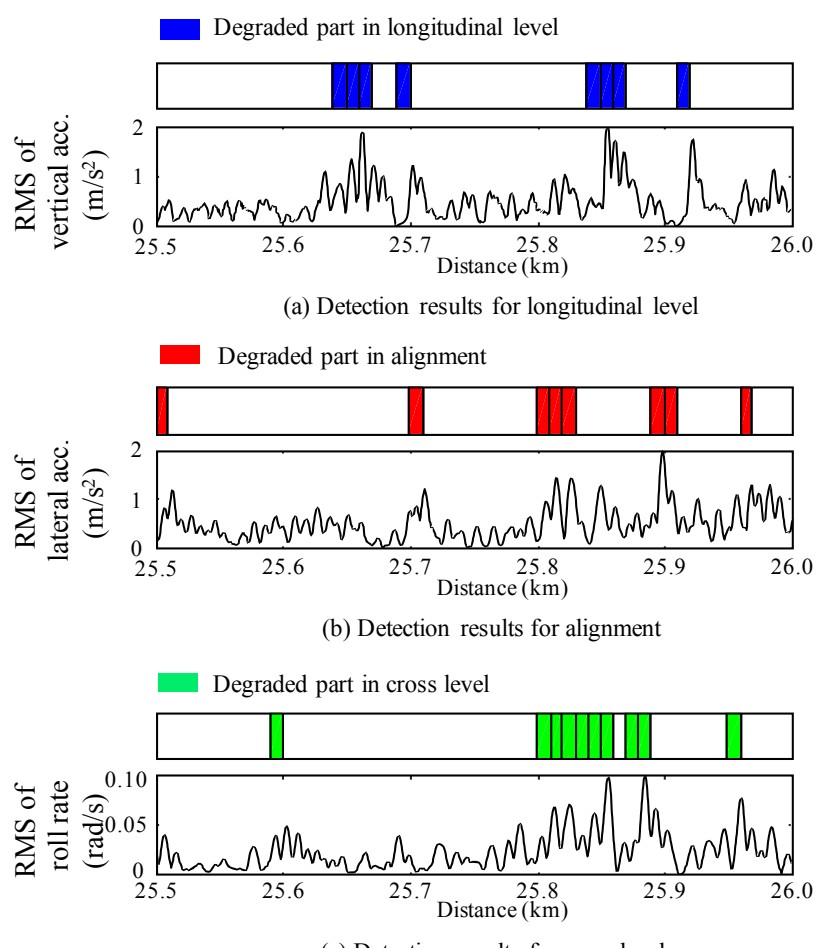

**Figure 14.** Detection results for regional railway line. (**a**) Degraded part in longitudinal level. (**b**) Degraded part in alignment. (**c**) Degraded part in cross level.

## 7. Conclusions

In this study, we analysed the effects of railway track irregularities on car-body vibrations through railway vehicle travel simulations conducted by using the SIMPACK software package. We subsequently developed an algorithm based on machine learning to diagnose track conditions by extracting features to detect track irregularities.

We used the developed algorithm to detect track faults from the car-body vibrations generated by the simulation. Additionally, we examined the possibility of diagnosing track conditions by using real-world data collected by measurements on an actual regional railway line. The results of this study indicated that the developed algorithm could automatically detect faults associated with longitudinal level, alignment, and cross level irregularities by using measured car-body vibrations. In future work, we plan to create higher-precision training data to improve detection performance. We will also attempt to increase the effectiveness of the current approach by increasing the number of training data clusters and developing an algorithm that can detect the magnitude of track irregularities.

**Funding:** This research was funded by JSPS KAKENHI Grant Number 17K06240.

**Acknowledgments:** We would like to thank Editage (www.editage.jp) for English language editing.

**Conflicts of Interest:** The author declares no conflict of interest.

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
