# Peer review of "Condition Monitoring of Railway Tracks from Car-Body Vibration Using a Machine Learning Technique"

_applsci, doi:10.3390/app9132734_

Reviewer 1 Report

The paper is well written and the contents is very useful for railway industry.

I have a short comment on the detection of track irregularity.

In this simulation and examples with an in-service train, the evaluated track seem to be a straight.

It seems that alignment irregularity may change in accordance with the radii of curved tracks.

If the effect is negligible, I recommend you to comment on it with a short sentence in the paper.

Author Response

In this simulation and examples with an in-service train, the evaluated track seem to be a straight. It seems that alignment irregularity may change in accordance with the radii of curved tracks. If the effect is negligible, I recommend you to comment on it with a short sentence in the paper.

Thank you for your valuable comments on this paper. 

I agree that alignment irregularity is affected by the curve radius. We have not checked the effect so far but we need to study as the future work.

We added a following sentence for explain the current situation in the end of section 4.2.

“This simulation study is carried out based on the straight track. Alignment irregularity in curved track is not considered in this paper but should be considered in the next work.”

Reviewer 2 Report

The manuscript 'Condition monitoring of railway track from car-body vibration' describes a railway on-board system for the detection of track irregularities. The system uses machine learning techniques to identify the track condition by integrating a generic train model built with a commercial software.

The manuscript is well written and it can have an impact on the practice of identifying track condition in the field of railway engineering. The theoretical background of the methods used for the implementation of the track condition monitoring system is weakly presented or not present from the manuscript, however, the study includes a real-life measurement example. Given the above considerations, the reviewer can recommend the publication of the manuscript in an application based journal such the Applied Sciences.

Author Response

The manuscript is well written and it can have an impact on the practice of identifying track condition in the field of railway engineering. The theoretical background of the methods used for the implementation of the track condition monitoring system is weakly presented or not present from the manuscript, however, the study includes a real-life measurement example. Given the above considerations, the reviewer can recommend the publication of the manuscript in an application based journal such the Applied Sciences.

Thank you for your valuable comments on this paper. The theoretical background for the track condition monitoring system is not described enough on this paper because we have already reported in published papers.

Reviewer 3 Report

It is an interesting paper and the reviewer think it’s worth for publication after addressing the following suggestions.

Would t the system get influence by faulty train (e.g. wheel flat)?

Line 120-121, the authors mentioned there is a high correlation between the car-body vibration and track irregularities. Can authors show some evidences of these?

Line 130-131, the authors compared the results from normal and increases track irregularities. Can the author explain a bit more specific about how they apply track irregularities and worse irregularities?

Based on N=26 with sampling frequency 82 Hz and train speed 60 km/hr mentioned in line 140, the track should be diagnosed for around every 5.3 m. Why in line 160, the authors mentioned the track is diagnosed for every 10 m?

Author Response

It is an interesting paper and the reviewer think it’s worth for publication after addressing the following suggestions.

Would t the system get influence by faulty train (e.g. wheel flat)?

Thank you for your valuable comments on this paper. 

The effect of vehicle faults such as wheel flat will appear on the measured acceleration. It is possible to identify the possibility of the faulty train by looking at the measurement data of different places. If the measurement data increases at many places, it should be the effect of the faulty train. In that case, we need to change the measurement train but we have no experience so far. We added a following sentence for further explanation in the end of section 5.2.

“The effect of vehicle faults such as wheel flat will appear on the car-body vibration. It is possible to identify the possibility of the faulty train by looking at the measurement data of different places.”

Line 120-121, the authors mentioned there is a high correlation between the car-body vibration and track irregularities. Can authors show some evidences of these?

We added two references for the evidences.

Line 124: “Because a high degree of correlation exists between these track irregularities and car-body vibrations, the magnitude of car-body vibrations can be an effective method to determine track conditions [12,13].”

Reference

[12] Vinberg, E.; Martin, M.; Firdaus, A.; Tang, Y. In Railway Applications of Condition Monitoring; KTH Royal Institute of Technology, DOI: 10.13140/RG.2.2.35912.62729, 2018

[13] Kairas, T.; Berg, M.; Stichel, S.; Li, M.; Thomas, D.; Dirks, B. Correlation of track irregularities and vehicle responses based on measured data. Vehicle System Dynamics 2018, 56:6, 967–981.

Line 130-131, the authors compared the results from normal and increases track irregularities. Can the author explain a bit more specific about how they apply track irregularities and worse irregularities?

We added in section 4.3:

“Degraded track is generated by the power spectral density of track where the standard deviation becomes twice as large as the normal track.”

Based on N=26 with sampling frequency 82 Hz and train speed 60 km/hr mentioned in line 140, the track should be diagnosed for around every 5.3 m. Why in line 160, the authors mentioned the track is diagnosed for every 10 m?

We diagnose track for every 10m under the consideration of GNSS error. Sliding window size N=26 was wrong and now changed to N=4.

Now we corrected as: Line 140: “We compute RMS values by using N = 4 and a sampling frequency of 82 [Hz].”

Line 153: “The section length 10m is decided based on the consideration of GNSS error.”

Reviewer 4 Report

See attached file.

Author Response

The paper is well written. The issue also addressed here is interesting. The numerical result has been presented by several figures. But the author should be consider about the optimization procedure for the hyperparameter of Kernel see : Tran, K. P., Huong, T. T. (2017, October). Data driven hyperparameter optimization of one-class support vector machines for anomaly detection in wireless sensor networks. In 2017 International Conference on Advanced Technologies for Communications (ATC) (pp. 6-10). IEEE. for example.

Thank you for your valuable comments on this paper.

We added in section 4.4: “We have not considered for optimising parameters in SVD [14] in this study, but it should be considered in the future work.”

Reference

 [14] Tran, K. P.; Huong, T. T.; Data driven hyperparameter optimization of one-class support vector machines for anomaly detection in wireless sensor networks. 2017 International Conference on Advanced Technologies for Communications (ATC)2017. 6–10.

Some minor corrections:

1.Title: Does it clearly describe the article ?

Now we changed the title as: 

“Condition monitoring of railway track from car-body vibration using machine learning technique”

2.Page 5: The formula (1) of root mean square (RMS) need to be clarified.

We added:

“xj  is the measured vertical acceleration, lateral acceleration or roll angular velocity of car-body.”

3.The author could add some comments for the result of classifying in section 4.4.

We added:

Line 169: “We have not considered for optimising parameters in SVD [14] in this study, but it should be considered in the future work.”